# Thrust: Adaptively Propels Large Language Models with External Knowledge

**Xinran Zhao**[1,2]   **Hongming Zhang**[1]   **Xiaoman Pan**[1]   **Wenlin Yao**[1]
**Dong Yu**[1]   **Jianshu Chen**[1]
[1]Tencent AI Lab, Bellevue, [2]Language Technologies Institute, Carnegie Mellon University

## Abstract

Although large-scale pre-trained language models (PTLMs) are shown to encode rich knowledge in their model parameters, the inherent knowledge in PTLMs can be opaque or static, making external knowledge necessary. However, the existing information retrieval techniques could be costly and may even introduce noisy and sometimes misleading knowledge. To address these challenges, we propose the instance-level adaptive propulsion of external knowledge (IAPEK), where we only conduct the retrieval when necessary. To achieve this goal, we propose measuring whether a PTLM contains enough knowledge to solve an instance with a novel metric, ***Thrust***, which leverages the representation distribution of a small number of seen instances. Extensive experiments demonstrate that ***Thrust*** is a good measurement of PTLM models' instance-level knowledgeability. Moreover, we can achieve higher cost-efficiency with ***Thrust*** score as the retrieval indicator than the naive usage of external knowledge on 88% of the evaluated tasks with 26% average performance improvement. Such findings shed light on the real-world practice of knowledge-enhanced LMs with a limited knowledge-seeking budget due to computation latency or costs [*] .

## 1 Introduction

Knowledge is crucial for understanding human language and solving various NLP tasks [59]. In recent years, the pre-trained language models (PTLM) have demonstrated great success on various NLP tasks [10, 43, 32, 44, 5] by storing rich encyclopedic [42] and commonsense [25] knowledge in their model parameters. However, such implicit knowledge could be opaque, static, or inefficient [23]. These issues motivate the common practice of seeking external knowledge [30, 57, 53, 17] with information retrieval methods and augmenting the inference models (e.g., PTLMs) [20, 12, 24] with the retrieved knowledge.

However, this approach has two limitations: (i) extracting external knowledge with existing information retrieval tools can be costly for a large-scale knowledge resource. (ii) external knowledge can be unnecessary or even misleading. For instance, one of the best retrieving models ColBERT v2 [46] achieved 68.9 Success@5 on Natural Question [27], which suggests that gold documents do not appear in the top five retrieved documents for 31.1% of the queries. Considering the limited input sequence length, the most useful documents may not be included for generating a prediction, while others may add noise to the model. On the other hand, PTLMs, which grow from millions (e.g., BERT [10]) to billions of parameters (e.g., OPT [61]), may solve the queries directly without

---

[*]Work done during interning at Tencent AI Lab, Bellevue. Corresponding contact email addresses: `xinranz3@andrew.cmu.edu`, `{hongmingzhang,xiaomanpan,wenlinyao,dyu,jianshuchen}` `@global.tencent.com`. Our code is available at `https://github.com/colinzhaoust/thrust_` `neurips2023`.

37th Conference on Neural Information Processing Systems (NeurIPS 2023).

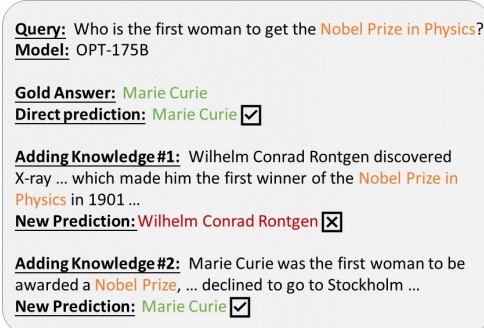

Figure 1: The predictions by OPT-175B without/with external knowledge retrieved via DPR [20] from Wikipedia paragraphs. Although the top retrieved paragraphs are relevant since the internal knowledge is already sufficient, the external knowledge can either be misleading (potentially due to the effect of *misprime* [21]) or less useful.

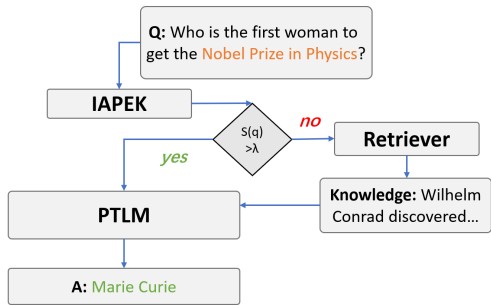

Figure 2: The pipeline of retrieval-augmented models with **IAPEK**. Unlike previous work (e.g., RAG [30]) where models directly seek for help from the retriever module, **IAPEK** module provides a confidence score $S(q)$ (e.g., ***Thrust***) on how well the PTLM can answer the question with internal knowledge and decides if the external retrieval is necessary.

external knowledge, making it unnecessary to seek external knowledge that signifies the noise issue. The instance shown in Figure 1 demonstrates the noise and inefficiency issues caused by external knowledge retrieval. OPT-175B can directly give the correct answer without external knowledge. However, with top external knowledge retrieved by DPR [20] from Wikipedia paragraphs, the external knowledge can be useless or even lead to wrong predictions.

The efficiency and noise issues motivate us to propose the **I**nstance-level **A**daptive **P**ropulsion of **E**xternal **K**nowledge (**IAPEK**), which adaptively retrieves external knowledge when it is necessary. We propose to measure whether a PTLM can solve a question instance with internal knowledge with a confidence score and reject using external knowledge when the confidence is high, instead of seeking retrieval directly for all the cases. The overall pipeline is shown in Figure 2. Specifically, we propose to solve this problem from the representation learning perspective based on two assumptions: (i) if a PTLM has mastered sufficient knowledge about a task, its hidden states should be able to cluster the samples from the task well enough; (ii) if the representation of a new instance deviates from these clusters in hidden states, this instance should be beyond the knowledge scope of the target PTLM. On top of these assumptions, we design a simple and lightweight metric ***Thrust*** to measure the distance between an instance's representation and the clusters of several observed examples in the same task.

To comprehensively understand the effectiveness of ***Thrust***, we conduct experiments on diverse NLP tasks. Experiments show that the average ***Thrust*** score of open-domain questions is significantly lower than other tasks, which aligns well with our intuition that open-domain QA tasks typically need external knowledge. These results indicate that ***Thrust*** is a good measurement of the models' knowledgeability. Extensive experiments also show that ***Thrust*** can improve the cost-efficiency of seeking and using external knowledge on 88% cases with 26% average performance improvement through identifying the instances that mostly require knowledge. We can also observe that, with ***Thrust***, we can achieve higher performance than injecting external knowledge for all instances, where models are benefited from both the performance and efficiency aspects. Such findings shed light on the real-world practice of knowledge-enhanced LMs with a limited budget for knowledge seeking due to computation latency or costs.

## 2 Approach

### 2.1 IAPEK: Problem Formulation

We begin with a formal problem definition of **I**nstance-level **A**daptive **P**ropulsion of **E**xternal **K**nowledge (**IAPEK**). For each instance of natural language query $q$ (e.g., a question in question-answering tasks), we first determine whether the PTLM has sufficient knowledge to solve the current problem. We achieve this by using a real-valued score function $s(q)$ that assigns a higher score to $q$ if the current PTLM has enough knowledge to solve the problem and vice versa. And we will retrieve

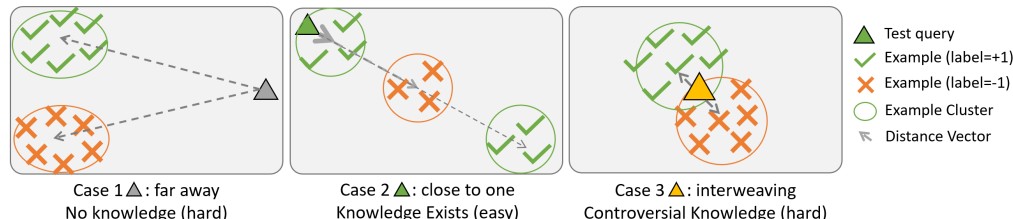

Case 1 ▲ : far away
No knowledge (hard)

Case 2 ▲ : close to one
Knowledge Exists (easy)

Case 3 ▲ : interweaving
Controversial Knowledge (hard)

▲ Test query
✓ Example (label=+1)
✗ Example (label=-1)
◯ Example Cluster
↖ Distance Vector

Figure 3: The intuition behind the proposed ***Thrust***, which are plotted in the hidden representation space of PTLM. We represent an incoming query instance by triangles and represent the instances used for constructing ***Thrust*** scores by ticks and crosses. In the *controversial* and *no knowledge* cases, the internal knowledge is insufficient to answer the query successfully, and external knowledge is needed to facilitate PTLM. In contrast, if the model finds the query close to one of the clusters, internal knowledge should be sufficient to solve the problem so that external knowledge is unnecessary.

external knowledge to facilitate PTLM once the score $s(q)$ falls below a threshold $\lambda$. By doing this, the PTLM can selectively retrieve external knowledge in an *instance-adaptive* manner and avoid unnecessary knowledge augmentation costs. To achieve such a mission, a critical step is to design an *instance-level* score function $s(q)$ that can effectively measure whether the PTLM has enough knowledge to solve the current particular input *instance*. In the remainder of the section, we will show how to construct such a function efficiently.

## 2.2 *Thrust*: Measuring the Knowledgeability of PTLMs

We now proceed to construct the scoring function $s(q)$ that measures the knowledgeability of a PTLM for solving an instance from a particular task $\mathcal{T}$. A seemingly straightforward approach is to adopt a supervised learning strategy to train such functional mapping from human-labeled data.

However, it is practically infeasible to manually annotate whether a given PTLM has sufficient knowledge at an instance level because the internal knowledge of PTLM is implicitly stored in its model parameters, which are hard to probe precisely for the IAPEK purpose. For this reason, we take an alternative approach by looking at the problem from the representation learning perspective. Specifically, our method is based on the following two assumptions. First, if a PTLM has mastered sufficient knowledge about a task, then its hidden states should be able to cluster the samples from the task well enough. For a classification task, samples from different classes should also be well separated in their (higher-level) hidden representations. Second, we further hypothesize that when a particular sample from the task deviates from these clusters in hidden states, this instance should be beyond the knowledge scope of the PTLM. In Figure 3, we illustrate the above intuitions for different cases. In other words, akin to the observations in [16, 28], we view the built-in knowledge of PTLM as a representational power that enables the deep models to learn more separable features. Based on such observations, we develop a knowledgeability scoring function for a PTLM by measuring how well it can separate samples in its hidden states.

We now proceed to design $s(q)$ that scores each query $q$ from a given downstream task $\mathcal{T}$. To begin with, we first collect a small set of samples from task $\mathcal{T}$ and compute their hidden state representations using the designated PTLM. (Empirically, we find that about 200 samples are sufficient in our experiments.) We denote such a representational mapping by a function $f(\cdot)$.[1] For generation tasks, we treat all instances as having a single dummy label. Next, we group these embedding vectors according to their class labels as $\mathcal{G}_l = \{(f(x_i), y_i) \mid y_i = l\}$, where $x_i$ and $y_i$ denote the $i$-th instance and its corresponding class label, respectively, and $l$ is the class index. We further cluster the samples in each $\mathcal{G}_l$ into $K$ clusters by applying the k-means algorithm to the vectors $f(x_i)$. For convenience, we introduce the notation $\mathcal{C}_{kl}$ to represent the set of samples in the $k$-th cluster of class $l$, and let $m_{kl}$ be the centroid vector corresponding to $\mathcal{C}_{kl}$.

---

[1] We use the last layer of hidden states as the embedding function. For T5-based models, we use the last layers of the decoders.

With the above notation, we define the **_Thrust_** score for a given query instance $q$ as:

$$s_{\text{thrust}}(q) \triangleq \left\| \frac{1}{N \cdot K} \sum_{l=1}^{N} \sum_{k=1}^{K} \frac{|\mathcal{C}_{kl}|}{\|d_{kl}(q)\|^2} \cdot \frac{d_{kl}(q)}{\|d_{kl}(q)\|} \right\|, \tag{1}$$

where $d_{kl}(q) \triangleq m_{kl} - f(q)$ is a vector pointing from $f(q)$ towards the centroid $m_{kl}$, $N$ is the number of classes, $K$ is the number of clusters per class, $|\mathcal{C}_{kl}|$ denotes the cardinality of the set $\mathcal{C}_{kl}$, and $\| \cdot \|$ denotes $\ell_2$-norm of a vector.

**The design principles**   Note that the expression inside the $\ell_2$-norm of (1) can be viewed as a weighted average of the normalized (unit) vectors $\{d_{kl}(q)/\|d_{kl}(q)\|\}$ that point from the query vector $f(q)$ towards the centroid vectors $\{m_{kl}\}$. The weighting is proportional to the cluster size and inversely proportional to the squared distance between the query and the centroid. Such a design choice is based on the following principles derived from the earlier representational assumption regarding knowledge. First, when samples from a task are well clustered and if $q$ is close to one of the clusters while being farther away from others, it means that the query instance $q$ can be well solved by the internal knowledge of PTLM and the thrust score should be high. Let $m_{kl}$ be the cluster centroid that $q$ is close to, then we observe that the corresponding $\|d_{kl}\|^2$ term in the denominator will make the corresponding term dominate and large in (1). Second, if $q$ is farther away from all the cluster centroids, i.e., the query is beyond the knowledge scope of the PTLM, then the quadratic term $\|d_{kl}\|^2$ would quickly suppress all the terms in (1), making the thrust score vanish. Third, when the PTLM cannot sufficiently cluster the task samples in its hidden states, it means that the PTLM does not have sufficient knowledge to solve the entire task. In this case, the unit vector $d_{kl}(q)/\|d_{kl}(q)\|$ would randomly point towards different directions so that the averaged vector inside the $\ell_2$-norm of (1) diminishes. Finally, the main reason that we first aggregate the samples within each class into $K$ clusters before computing the thrust score is that they may still be spread over multiple clusters even if they belong to the same class. The term $|\mathcal{C}_{kl}|$ in (1) is used to upweight the vectors $d_{kl}(q)/\|d_{kl}(q)\|$ that point to bigger clusters. Nevertheless, we find that $K$ can be relatively small.[2] In Section 4.1, we will conduct an experimental analysis to show that **_Thrust_** score designed in the above manner is a good measurement of a PTLM model's knowledgeability. In addition, we also carry out extensive ablation studies to examine these design choices (see Appendix).

**Practical considerations**   As we will show in Section 3, we only need about 200 samples from a task to form the clusters, and then we just need to store their corresponding centroids $m_{kl}$ (typically less than 50 vectors of dimension 300) for deployment. According to (1), computing the **_Thrust_** score for a query $q$ only needs to calculate the distance between $f(q)$ and these centroids, which takes about 0.001 second per query on average in our experiments (see Appendix). Therefore, **_Thrust_** is fairly lightweight and easy to be deployed in practical applications. The incurred extra computation complexity during the inference stage is $O(NK)$. Since $K$ and $N$ are generally small, this overhead is negligible compared to retrieving knowledge from a large external memory for each instance.

## 3   Experiment

In our experiments, PTLMs are examined under two settings. (i) **Zero-shot**. We consider T5 [44], GPT-J [54], OPT [61] and present queries with or without knowledge directly to the PTLMs. (ii) **Transfer-learning**. We use UnifiedQA [22] models that fine-tune T5 on multiple QA datasets.

After studying which model performs the best in utilizing external knowledge (the pre-condition of introducing retrievers), we then evaluate the cost-effectiveness of the proposed **IAPEK** with **_Thrust_** with the best performing models. We simulate the real-world usage scenario where we have limited bandwidth or budget to retrieve external knowledge. Specifically, we test on three scenarios based on the richness of available resources: **_scarce_** (25%), **_medium_** (50%), and **_abundant_** (75%). For example, **_scarce_** (25%) means that we set the threshold $\lambda$ (defined in Section 2.1) to 25 percentile of the scores of the 200 examples used to set up the clusters. As introduced in Section 2, we use **_Thrust_** to score and rank the instances by their need for external knowledge and select the instances that have a high demand for external knowledge (i.e., with low thrust scores). We compare the

---

[2]We choose $K$ to be $\max(\text{ceil}(\sqrt[4]{|D_{\mathcal{T}}^{\text{sample}}|}), 3)$, where $D^{\text{sample}}$ denotes the sample set of task $\mathcal{T}$.

performance between ***Thrust*** with two baselines, i.e., **IAPEK** (***Default***) that randomly samples X% (X=25, 50 or 75) examples to apply knowledge, and **IAPEK** (***BM25***) that ranks all instances by BM25 [52] and select top X% (X=25, 50 or 75) difficult examples.

We follow previous work to report accuracy for MC classification tasks. For open-domain QA, we report the QA-F1 score that measures the max uni-gram overlap between the model prediction and all gold answer candidates following previous work [22]. For the *without knowledge* setting, we directly pass the prompt-decorated question query to the model and select the choice with the highest probability as the answer. For the *with knowledge* setting, we append the knowledge piece after the prompt-decorated query and put "*Answer:* " at the end to pass to the models.

## 3.1 Datasets

We consider several knowledge-intensive tasks in our evaluation, i.e., Multiple-choice Classification (MC), which consists of seven datasets, and Open-domain QA, which consists of five tasks. Each data instance consists of **a query** (a piece of text containing a question or the sentences to be classified) and **an answer** (either the label words or the answers to the questions in the query).

Additionally, each instance may also contain a piece of potentially helpful **knowledge** for the query, which is either inherently relevant due to the task design, annotated by humans, or retrieved from Wikipedia paragraphs with DPR. Details of the datasets and corresponding external knowledge are as follows.

**Multiple-choice classification.** For MC classification, each query $q$ includes a sentence or a question and requires models to select the correct answer from a set of candidates. Specifically, (i) AG-News [62] asks the model to classify a piece of news into *political, sports, business,* or *technology*. We regard the titles of the news as the queries since they may already contain sufficient information and use the content of the news as the gold external knowledge. (ii) e-SNLI [6] is a natural language inference (NLI) task exploiting the role of explanations in textual entailment. Human-providing explanations are considered a strong source of external knowledge; (iii) StrategyQA [13] is a challenging multi-hop reasoning dataset that requires models to answer creative questions through strategical inference from implicit reasoning steps. We regard the original questions as queries and human-written explicit facts as external knowledge; (iv) CIKQA [60] is a commonsense inference task that combines pronoun coreference resolution, commonsense QA [51], COPA [45], and questions mined from ATOMIC knowledge graph [47]. We regard the original questions as queries and the supporting commonsense knowledge extracted from knowledge graphs (KGs) in the original work as the external knowledge; (v) BoolQ [8] contains encyclopedic questions that require models to answer *yes* or *no*. Following [22], we use the Wikipedia paragraphs retrieved by DPR as the external knowledge, which can be potentially noisy; (vi) ARC-E & ARC-C [9]: ARC is a challenging multiple-choice QA dataset that requires knowledge understanding and reasoning, which is partitioned to an Easy set (ARC-E) and a Challenge set (ARC-C), where the Challenge set questions are answered incorrectly by the retrieval-based or co-occurrence-based algorithms tested by the original authors. Similarly, we use the Wikipedia paragraphs retrieved by DPR as external knowledge.

**Open-domain QA.** For open-domain QA, each query $q$ contains an open question that typically requires solving an encyclopedic or commonsense inference. The generated answers can either be a few phrases or a single sentence. An example question is "*What does a drink from Narcissus's Spring cause the drinker to do*" and the expected answer generated from the language model is "*fall in love with themselves*". The involved datasets are HotpotQA [58], Natural Questions (NQ) [27], Web Questions [2], Curated TREC [1], and TriviaQA [19]. We use Wikipedia paragraphs retrieved by DPR as the external knowledge as a common practice [59], except for HotpotQA, where we use the passages that the queries are generated from as a gold knowledge resource.

The statistics of the involved datasets (e.g., query length and sizes of the splits) are reported in Appendix. We collect a benchmark with various datasets of different types, formats, and knowledge sources, where we will then evaluate the effectiveness of **IAPEK**. Some implementation details of each of the task are described in Appendix.

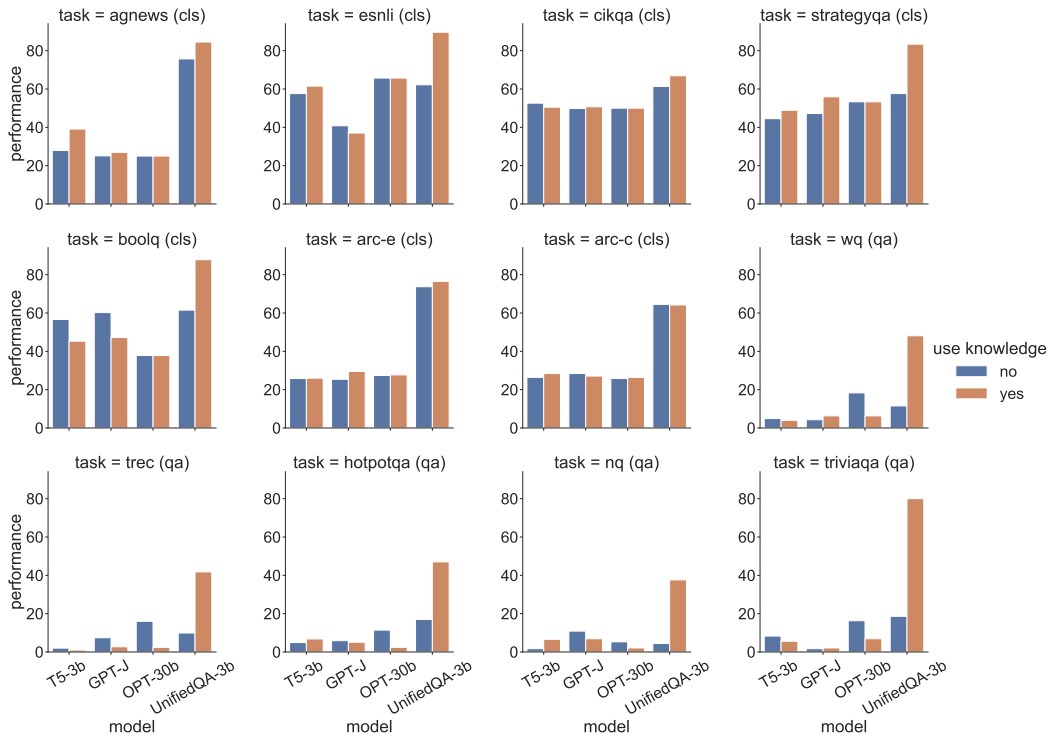

Figure 4: Performance of various models on MC classification tasks (accuracy) and open-domain QA tasks (QA-F1), denoted by (cls) and (qa), respectively. The x-axis represents the model names, which are shared across sub-figures. Use knowledge: yes or no denotes using full knowledge or not for all queries. UnifiedQA denotes T5 models with different sizes fine-tuned on the UnifiedQA dataset.

## 3.2 Performance of Using External Knowledge

Figure 4 presents the model performance on both the MC classification and open-domain QA tasks[3]. For the MC classification tasks, we can observe that for the zero-shot setting (T5-X, GPT-J, and OPT), models do not consistently benefit from external knowledge. In addition, the ability to utilize external knowledge is also not improved as the parameter size grows, which indicates that simply using larger models may not be the solution for better using the knowledge. For the transfer-learning setting (UnifiedQA-X), although *AGNews*, *e-SNLI*, *CIKQA*, and *StrategyQA* are not seen during the training of UnifiedQA models, we can observe that models achieve better performance than vanilla T5 models at different sizes. Under the *with knowledge* case, the UnifiedQA models achieve significant improvement for utilizing external knowledge compared to the zero-shot models, *UnifiedQA-3b* achieves the best performance on all the tasks, which indicates that models can learn and transfer the ability to utilize external knowledge with instances containing external knowledge. Moreover, for open-domain QA tasks, we can see models (T5, GPT-J, OPT) get no benefits from external knowledge in 11 out of 25 cases under the zero-shot setting, while UnifiedQA models achieve significant performance gain after adding external knowledge. External knowledge may introduce extra noise if the model does not learn to utilize knowledge, which indicates the importance of instructing PTLMs to learn how to use knowledge through second-stage fine-tuning.

In conclusion, we find that fine-tuning with instances containing external knowledge is an effective way to help models learn to use external knowledge. Since the pre-condition of using IAPEK is that the model can utilize external knowledge well, we conduct experiments with UnifiedQA only when evaluating the performance of ***Thrust***.

---

[3]the detailed numeric values, design choice ablation, and known limitations are presented in Appendix.

Table 1: Performance of **IAPEK** based on *Thrust*. As defined in Section 3, performances of **Default**/**Thrust** are presented before/after the vertical bar for **scarce**, **medium**, and **abundant** cases. If performance increases with **Thrust**, the score will be marked in green and otherwise in red. WQ and TREC denote the tasks of Web Questions and Curated TREC, respectively.

| Dataset | UnifiedQA-base | | | UnifiedQA-large | | | UnifiedQA-3b | | |
|---|---|---|---|---|---|---|---|---|---|
| | scarce | medium | abundant | scarce | medium | abundant | scarce | medium | abundant |
| AGNews | 50.7 \| 55.6 | 52.8 \| 56.3 | 55.0 \| 56.8 | 70.2 \| 69.1 | 69.4 \| 70.2 | 68.7 \| 70.6 | 77.9 \| 78.4 | 80.1 \| 80.4 | 82.3 \| 82.3 |
| e-SNLI | 46.5 \| 66.6 | 54.4 \| 68.3 | 62.3 \| 69.6 | 50.7 \| 71.1 | 58.5 \| 72.2 | 66.4 \| 73.2 | 69.1 \| 86.3 | 75.9 \| 87.5 | 82.8 \| 88.8 |
| CIKQA | 56.9 \| 59.6 | 57.8 \| 59.6 | 58.7 \| 59.9 | 60.2 \| 62.1 | 60.8 \| 62.3 | 61.5 \| 62.4 | 62.7 \| 66.9 | 64.1 \| 66.9 | 65.5 \| 66.9 |
| StrategyQA | 50.7 \| 55.6 | 52.8 \| 56.3 | 55.0 \| 56.8 | 52.9 \| 62.1 | 57.4 \| 65.3 | 61.9 \| 65.9 | 64.1 \| 74.3 | 70.5 \| 81.4 | 77.0 \| 82.9 |
| BoolQ | 65.5 \| 76.2 | 70.7 \| 79.9 | 75.8 \| 80.9 | 65.9 \| 77.7 | 72.1 \| 81.3 | 78.3 \| 84.4 | 68.1 \| 79.1 | 74.6 \| 85.7 | 81.2 \| 87.1 |
| ARC-E | 50.7 \| 55.6 | 52.8 \| 56.3 | 55.0 \| 56.8 | 64.5 \| 64.6 | 65.0 \| 64.7 | 65.5 \| 65.1 | 74.4 \| 74.6 | 75.1 \| 74.9 | 75.8 \| 75.1 |
| ARC-C | 44.9 \| 43.8 | 45.0 \| 44.5 | 45.1 \| 44.8 | 53.8 \| 50.8 | 52.3 \| 51.2 | 50.9 \| 51.5 | 64.5 \| 63.9 | 64.4 \| 64.9 | 64.3 \| 65.6 |
| WQ | 19.2 \| 26.3 | 27.5 \| 42.1 | 35.8 \| 43.8 | 22.5 \| 38.5 | 30.5 \| 39.0 | 38.5 \| 46.0 | 20.9 \| 19.3 | 30.0 \| 35.4 | 39.1 \| 46.4 |
| TREC | 13.5 \| 33.6 | 21.3 \| 36.4 | 29.1 \| 36.9 | 30.8 \| 32.7 | 32.7 \| 36.0 | 34.6 \| 36.3 | 19.6 \| 37.8 | 27.0 \| 40.6 | 34.4 \| 40.9 |
| HotpotQA | 25.2 \| 32.9 | 30.2 \| 35.5 | 35.2 \| 37.8 | 26.7 \| 35.2 | 32.1 \| 37.5 | 37.4 \| 40.2 | 24.9 \| 41.9 | 32.3 \| 43.9 | 39.7 \| 45.7 |
| TriviaQA | 32.0 \| 52.7 | 43.2 \| 56.4 | 54.4 \| 60.0 | 32.4 \| 59.7 | 46.4 \| 64.3 | 60.5 \| 71.8 | 39.2 \| 68.3 | 52.8 \| 71.0 | 66.4 \| 73.4 |
| NQ | 20.0 \| 33.0 | 24.9 \| 33.5 | 29.7 \| 33.9 | 12.0 \| 34.8 | 20.1 \| 35.2 | 28.2 \| 35.7 | 12.8 \| 35.9 | 21.1 \| 36.5 | 29.4 \| 37.0 |

## 3.3 Performance of *Thrust*

Table 1 shows the results of UnifiedQA after adding knowledge to X% (X=25, 50 or 75) that needs knowledge most according to our **Thrust** score. We can see that **Thrust** consistently contributes to the performance from the base to the 3B model. Through clustering the instances, we acquire the whole instance distribution in the eyes of the models. Then with distance to the cluster, **Thrust** represents how well the model can categorize a new query vector and find its similarity with others on the task. Leveraging such information, **Thrust** identifies the *no knowledge* and *controversial knowledge* cases well and puts the knowledge into the most necessary ones.

Additionally, the gain is higher when the portion of augmented instances is smaller. For instance, for UnifiedQA-3b, the gains from **Thrust** with the **scarce** case are 6.1%, 13.56% on MC classification and QA tasks, respectively, while for the **abundant** case, the gains are 2.8% and 6.8%. Such observation shows that **Thrust** is most effective in identifying the most necessary cases. One potential reason is that **Thrust** is sensitive to the distance change, so the isolated instances (*no knowledge* case in Figure 3) can be easily identified. With **Thrust** the performance of cases using fewer resources can sometimes surpass that with high resource requirement for **Default**, e.g., for UnifiedQA-base on all classification tasks except ARC-C, **Thrust**(scarce) performs better than **Default**(medium). Interestingly, we also observe consistent failure cases on ARC-C. This is because the queries are designed as open questions, and the answers are usually about plans or opinions instead of facts. Thus, it is hard for small models to extract useful information from Wikipedia documents.

## 4 Analysis

### 4.1 Primary Study: Distribution of *Thrust* scores

To investigate whether **Thrust** is a good measurement of models' knowledgeability, we plot in Figure 5 the distribution of **Thrust** scores with the strongest inference model evaluated (i.e., UnifiedQA-3b)[4]. We use Kernel Density Estimation to smooth the distribution. From the distribution, we can see that low **Thrust** scores (i.e., the query needs external knowledge) frequently appear in most query instances of the open-domain QA tasks such as *HotpotQA*, *TriviaQA* and *NQ*, which are designed to solve with external knowledge. On the other hand, for *e-SNLI* and *BoolQ*, external knowledge is not always necessary, which is consistent with the design purpose of these tasks. To conclude, by correctly predicting whether a certain task needs external knowledge (e.g., open-domain QA tasks), **Thrust** score is shown to be a good measurement of a PTLM model's knowledgeability.

---

[4]For a clear presentation, we only include representative tasks directly relevant to knowledge and reasoning from each category. The selected tasks are: *e-SNLI*, *BoolQ*, *CIKQA*, *HotpotQA*, *TriviaQA*, and *NQ*. The distribution of all tasks can be found in Appendix, where the trend is consistent. In visualization, the score can be lower than zero due to smoothing. The real scores will always be greater than zero.

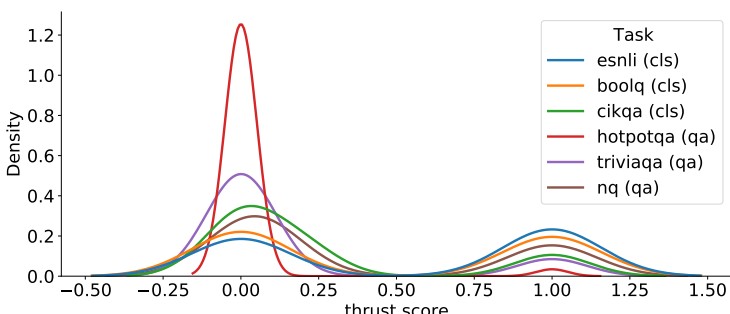

Figure 5: Distribution of *Thrust* scores for various tasks by using UnifiedQA-3b to create the hidden representations. Kernel Density Estimation is used to smooth the distributions. Low scores imply that the instances are less likely to be solved with internal knowledge and vice versa. *Thrust* scores predict that most query instances from open-domain QA tasks require external knowledge while others need less. The results are consistent with the original design purposes of these tasks.

Table 2: Performance of **IAPEK** for UnifiedQA-3b based on *Thrust* and BM25. With *scarce*, *medium*, and *abundant*, performances of **Default/BM25** and **Default/*Thrust*** are presented before/after the vertical bar. If performance increases with *Thrust*, the score will be marked in green, otherwise red.

| Dataset | BM25 | | | Thrust | | |
|---|---|---|---|---|---|---|
| | scarce | medium | abundant | scarce | medium | abundant |
| AGNews | 77.9 \| 77.0 | 80.1 \| 79.2 | 82.3 \| 81.3 | 77.9 \| 78.4 | 80.1 \| 80.4 | 82.3 \| 82.3 |
| e-SNLI | 69.1 \| 68.4 | 75.9 \| 75.6 | 82.8 \| 83.0 | 69.1 \| 86.3 | 75.9 \| 87.5 | 82.8 \| 88.8 |
| CIKQA | 62.7 \| 62.3 | 64.1 \| 64.4 | 65.5 \| 66.1 | 62.7 \| 66.9 | 64.1 \| 66.9 | 65.5 \| 66.9 |
| StrategyQA | 64.1 \| 63.3 | 70.5 \| 68.3 | 77.0 \| 78.1 | 64.1 \| 74.3 | 70.5 \| 81.4 | 77.0 \| 82.9 |
| BoolQ | 68.1 \| 68.4 | 74.6 \| 75.9 | 81.2 \| 82.0 | 68.1 \| 79.1 | 74.6 \| 85.7 | 81.2 \| 87.1 |
| ARC-E | 74.4 \| 74.9 | 75.1 \| 75.3 | 75.8 \| 76.3 | 74.4 \| 74.6 | 75.1 \| 74.9 | 75.8 \| 75.1 |
| ARC-C | 64.5 \| 65.2 | 64.4 \| 66.2 | 64.3 \| 66.6 | 64.5 \| 63.9 | 64.4 \| 64.9 | 64.3 \| 65.6 |
| WQ | 20.9 \| 19.0 | 30.0 \| 28.2 | 39.1 \| 37.3 | 20.9 \| 19.3 | 30.0 \| 35.4 | 39.1 \| 46.4 |
| TREC | 19.6 \| 20.4 | 27.0 \| 28.1 | 34.4 \| 36.3 | 19.6 \| 37.8 | 27.0 \| 40.6 | 34.4 \| 40.9 |
| HotpotQA | 24.9 \| 25.2 | 32.3 \| 32.8 | 39.7 \| 40.6 | 24.9 \| 41.9 | 32.3 \| 43.9 | 39.7 \| 45.7 |
| TriviaQA | 39.2 \| 34.2 | 52.8 \| 50.0 | 66.4 \| 65.4 | 39.2 \| 68.3 | 52.8 \| 71.0 | 66.4 \| 73.4 |
| NQ | 12.8 \| 12.9 | 21.1 \| 21.1 | 29.4 \| 29.6 | 12.8 \| 35.9 | 21.1 \| 36.5 | 29.4 \| 37.0 |

## 4.2 IAPEK-Thrust versus IAPEK-BM25

We use BM25 [52], a common approach to evaluating the difficulty of queries, as an alternative to *Thrust* to perform **IAPEK**. Specifically, we regard each test input as the query and all training data input as the corpus to extract the score. We use the average relevance score across the corpus to rank each test input. From Table 2, We can observe **IAPEK** performs well with BM25 as the difficulty score on QA tasks. Except WQ and NQ, we observe better performance (marked in green) than the default setting. However, *Thrust* shows larger (e.g., for QA tasks) and more robust improvement (e.g., for classification tasks) than the BM25 baseline.

## 4.3 Layer ablation

Since we cast instances into the representation space, a crucial factor for *Thrust* is the layer of the PTLMs to use. To investigate the effect of using different layers, we conduct experiments on UnifiedQA-3b with the same setting as in Section 3. Figure 6 presents the performance of adding 25%, 50%, and 75% knowledge-augmented instances with *Thrust* with hidden states of different layers. We can observe that, for most tasks, there is no significant difference across layers, which shows the robustness of *Thrust* and potential to accelerate the computation by using early layers. However, for *StrategyQA* and *Web Questions*, the middle-layer representation may worsen the overall performance. One possible reason is that early layers in the model contain rich semantic information, and later layers contain task-specific information [33], so both can act as good representations of the instances. However, in the middle layers, rich semantic features are abandoned during extracting task-specific features.

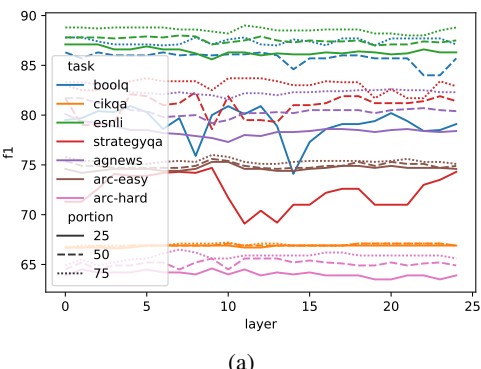
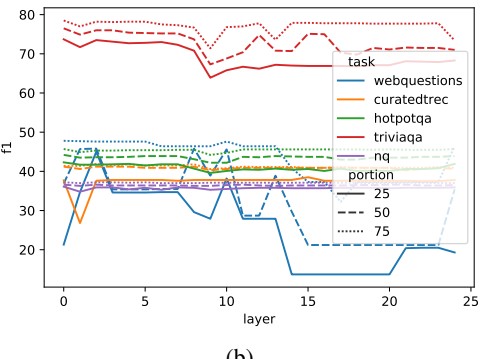

|(a)|(b)|

Figure 6: Layer-wise ablation across tasks and portions of instances augmented with knowledge for (a) MC classification tasks and (b) open-domain QA tasks. The x-axis denotes the layer index of Unified-3b decoder that is used to obtain the hidden representations (i.e., $f(\cdot)$ in (1)). For most tasks, the results are not sensitive to the specific layer index. For some tasks (e.g., StrategyQA), choosing middle layers for representation slightly degrades the performance.

Table 3: Comparison between IAPEK-***Thrust*** and the costly full knowledge usage based on UnifiedQA models of different sizes. 99% Full denotes 99% performance of models with full knowledge. The knowledge type is noted in brackets, where *g* denotes gold knowledge, *h* denotes human annotations, and *r* denotes the knowledge retrieved from Wikipedia or knowledge graphs. WQ and TREC stand for Web Questions and Curated TREC, respectively. If ***Thrust*** achieves better performance than using full knowledge, we mark the entry with *.

| Size | ***Thrust***>99% Full | | | | ***Thrust***<99% Full | | | |
|------|------|------|------|------|------|------|------|------|
| base | BoolQ(r)* ARC-E(r) | CIKQA(r)* ARC-C(r)* | AGNews(g)* WQ(r) | StrategyQA(h)* TriviaQA(r)* | e-SNLI(h) | TREC(r) | HotpotQA(g) | NQ(r) |
| 3b | BoolQ(r)* ARC-E(r) | CIKQA(r)* ARC-C(r)* | AGNews(g) TriviaQA(r) | StrategyQA(h) HotpotQA(g)* | e-SNLI(h) | TREC(r) | WQ(r) | NQ(r) |

## 4.4 Comparison with Full Knowledge Usage

We denote simply using external knowledge for all instances as a costly but straightforward way of leveraging external knowledge. Since the big models might be sufficient for certain instances and the external knowledge might introduce extra noise, ***Thrust*** can help identify instances requiring (or not) knowledge and achieves higher overall performance on the whole dataset compared to seeking and adding knowledge indiscriminately. Table 3 presents the comparison between adaptive and indiscriminate knowledge propulsion. ***Thrust*** here denotes the best performance achieved when less than 90% of instances use external knowledge. We can observe that, for 2/3 tasks for UnifiedQA-base and UnifiedQA-3b, ***Thrust*** achieves better performance or less than 1% drop than using knowledge for all instances. Such results illustrate that ***Thrust*** can help avoids potential noise. On the other hand, we can also observe that for *e-SNLI*, *TREC*, and *NQ*, the full knowledge setting performs better than ***Thrust***. It means retrieving and adding high-quality knowledge always benefits the models as long as they have been fine-tuned to know the usage of external knowledge.

## 5 Related Work

**PTLM with external knowledge.** The paradigm of retrieving knowledge from knowledge bases, augmenting PTLMs, and solving downstream tasks has been widely explored in the community of NLP [29, 4, 15, 39, 48]. The knowledge bases can range from knowledge graphs [57], documents [40], pre-processed vectors [53], other PTLMs [49], search engines [34], to Wikipedia documents as used in this work. To augment the PTLMs, common practice includes creating synthesizing datasets [56], adding knowledge to the prompts [55, 36], create demonstrations [5], and extending feature vectors [24]. The contribution of **IAPEK** is orthogonal to the above work since it presents a gated

framework to reject external annotations or retrieval that can be extended to the above frameworks. The extension is lightweight since ***Thrust*** requires queries only, not labels nor gold answers.

**Hardness and Confidence Estimation in PTLMs.** Much previous work studies the estimation of dataset hardness and model confidence under the context of PTLMs. For dataset hardness, previous research discovers using the cumulative area under the loss curves (RDA [41]), entropy difference between the trial and null cases ($\mathcal{V}$-Usable information [11]), and variance of losses of neighbor sentences (Sensitivity Measurement [14]) These methods achieve great correlation with the model performance when analyzing the test set. However, the test set labels are required and can not be applied when predicting the answers. Another line of work focuses on estimating the expected calibration errors (ECE) for classification [26], QA [18], and math [31] datasets, as a reflection of model certainty on the correct answers. ECE can be considered an orthogonal evaluation metric to measure the model's capability of understanding the tasks, compared to common metrics such as accuracy.

Most previous work can be considered a posterior analysis of the model capability. In this work, instead, we estimate the pragmatic confidence at the test time to empirically increase the performance with a limited budget or bandwidth to acquire knowledge.

## 6 Discussion

### 6.1 Extended Usage

We expect the idea of adaptive knowledge injection to be extendable beyond QA and MC questions, such as trail prediction tasks including ECBD [37] or EKP [38]. By design, Thrust is independent of the type of external knowledge, so that the adaptively used external knowledge can be any sort, for example, definitions in EKP. Furthermore, Thrust can also be used as a way to measure the expected performance without extensive fine-tuning, as shown in Table 5 in the appendix.

On the other hand, we present our pioneer study on instruction fine-tuned model Flan-T5 [7] in Section A.4 in the appendix and show that ***Thrust*** performs better on CIKQA with Flan-T5 than vanilla T5, where the best performance is achieved with 40% examples not using external knowledge.

We also believe future work can be done on how ***Thrust*** collaborates with different kinds of prompts to include the knowledge or prefix tuning. We discuss the limitations of our current design (i.e., cold start, assuming white-box language models, and extendability on other kinds of retrieval argumentation) in Section A.1 in the appendix.

### 6.2 Time Sensitivity

We regard time sensitivity as a part that can be done in the **IAPEK** framework, but not by ***Thrust***, as the framework is motivated by both noise and staticity issues. Another orthogonal kind of score measuring time sensitivity can be designed to decide if updated knowledge retrieval is necessary, for example, based on [35].

## 7 Conclusion

In this work, we propose **I**nstance-level **A**daptive **P**ropulsion of **E**xternal **K**nowledge (**IAPEK**) as a solution to propel model performance with external knowledge. Accordingly, we propose a simple and effective instance-level metric, ***Thrust***, to perform the adaptive knowledge injection. Extensive experiments show that ***Thrust*** is a good indicator of models' knowledgeability and can improve the performance of utilizing external knowledge under various settings. Understanding the delicate usage of potentially noisy knowledge for PTLMs can further enable the models to conduct inference beyond the limitation of internal knowledge.

## 8 Acknowledgements

We are grateful to Sherry Tongshuang Wu, Shikhar Murty, Zhengxuan Wu, Ashwin Paranjape, John Hewitt, Christopher Manning, Eric Mitchell, Sihao Chen, Ben Zhou, Tong Chen, and the anonymous reviewers for their helpful and insightful comments.

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

# A Appendix

## A.1 Limitations

**Cold Start.** In the ideal case, a module distinguishes if a query requires external can do it in a zero-shot manner. However, as we show in Section 4.1 of our paper, we practically find that the distribution of *Thrust* scores of various tasks can be very different due to the essence of the task collection and the type of external knowledge needed. Considering such an effect, we need 200 examples to estimate the clusters needed to set up the computation, which is still lightweight in real-world scenarios. In the future, we will explore the usage of meta-learning to allow a few-shot start or even a cold start of IAPEK.

**Black-box LLM**: At the current stage, our model can work with first-layer or last-layer representations (as discussed in Figure 6 of our submitted paper), which are provided by some black-box models. To adapt to completely black-box LLMs, there are two ways we can think at this stage: (1) similar to Black-Box Tuning [50], we use the prompt embeddings adjusted by the derivative-free optimizer optimized over the black-box model outputs as our representation; (2) we use original or distilled smaller models from the same family to acquire representation (e.g., original GPT-2 or GPT-2 fine-tuned by a set of query and answers from GPT-4). We experimented with using T5-base representation to conduct IAPEK for T5-large models. It showed slightly worse but not completely ruined performance.

**Extension to other Retrieval-augmented Models.** In this paper, we propose a new module for the pipeline of retrieval-augmented models. We first comprehensively examined if and how external knowledge is useful with language models. Next, we examine the performance of the module **IAPEK** with the lightweight *Thrust* score we define as a potential implementation. We compare *Thrust* with BM25 with the default setting of retrieval augmented language models [17] and show its effectiveness. Since queries, not answers nor retrieved knowledge are required to set up *Thrust*, it can be applied to various other frameworks of retrieval augmented models [3]. However, it is beyond the scope of the project at the current stage, and the contribution of our module and the frameworks are orthogonal. We will extend *Thrust* to other retrieval-augmented models in future work.

## A.2 Implementation details

We conduct our experiments on a machine with 8 Nvidia P40 (24G) GPUs with CUDA 11 installed.

We use the Scikit-learn package [5] to measure the clusters with K-means and compute the distance between the query and cluster representations. The involved hyperparameters (including the number of clusters per class) are selected by Grid search on a smaller set of experiments. We initialize all parameters randomly or as the default of the Hugginface transformers package [6]. On average, each run of extracting the results for all the tasks under with/without knowledge cases takes around 20 hours. We run all experiments 3 times and report the averaged performance in the main content. For hyperparameters of the inference models, for the QA task, we set the maximum knowledge length as 480 tokens to ensure that query sentences stay in the input.

The generated answer for QA tasks for all the models is typically within 30 tokens. For classification tasks, for binary classification tasks (CIKQA, StrategyQA, BoolQ, and e-SNLI), we follow previous work to use "Yes or No?" as the suffix to the original query to guide the generative models. For AGNews, we use "political news, sports news, business news, and technology news" as the label words. We found that the default label word "word news" will largely degrade the performance of generative models on AGNews. We add "the news is about?" and provide the candidate categories as the suffix for AGNews. More details of our implementations can be found in the code attached.

## A.3 Dataset Details

The detailed statistics of the involved datasets are shown in Table 4. We sample 200 data points from each dataset to conduct the clustering step of *Thrust*. Difference datasets have different average query lengths and knowledge lengths due to the essence of the task creation and knowledge collection.

---

[5] https://scikit-learn.org/
[6] https://huggingface.co/

Table 4: Statistics of the selected datasets. Sample # denotes the number of examples used to calculate the clusters for **Thrust** scores. ARC-E and ARC-C denote the easy and hard ARC datasets. Q/K/A Len denotes the average number of words for the Queries/Knowledge/Answers, respectively.

| Dataset | Source | Sample # | Test # | Q Len | K Len | A Len |
|---|---|---|---|---|---|---|
| AGNews | gold | 200 | 7,600 | 8.1 | 35.9 | 1.0 |
| e-SNLI | human | 200 | 9,824 | 24.9 | 14.3 | 1.0 |
| StrategyQA | human | 200 | 229 | 10.8 | 33.5 | 1.0 |
| CIKQA | KG | 200 | 604 | 18.2 | 28.0 | 1.0 |
| BoolQ | retriever | 200 | 3,270 | 9.8 | 113.8 | 1.0 |
| ARC-E | retriever | 200 | 570 | 23.1 | 238.2 | 4.2 |
| ARC-C | retriever | 200 | 299 | 26.2 | 240.5 | 5.5 |
| HotpotQA | gold | 200 | 7,405 | 19.0 | 56.3 | 2.5 |
| NQ | retriever | 200 | 6,468 | 10.1 | 588.9 | 2.3 |
| Web Questions | retriever | 200 | 278 | 7.8 | 117.3 | 4.3 |
| Curated TREC | retriever | 200 | 116 | 8.4 | 116.5 | 7.7 |
| TriviaQA | retriever | 200 | 6,760 | 15.0 | 117.6 | 27.5 |

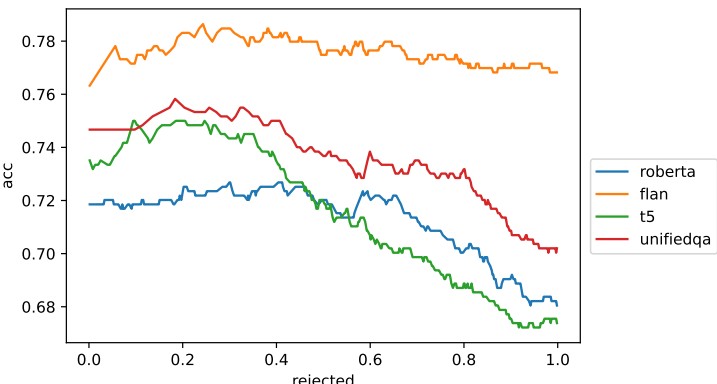

Figure 7: Performance of different models on CIKQA with different thresholds of **Thrust**. The X-axis denotes the portion of test examples that are selected to not use external knowledge. All model names denote the large versions of the model parameters.

Answer length 1 denotes tasks with yes and no answers. Otherwise, the answers with more than one token are either choices (for ARC-E and ARC-C) or free-form text sentences (for open-domain QA tasks). Examples of the dataset can be found in the attached data.

## A.4   Experiment with Flan-T5

Figure 7 presents the performance of **Thrust** on CIKQA with different models. From the figure, we can observe that  **Thrust**performs better with instruction fine-tuned Flan-T5 compared to the original T5 and UnifieedQA. With Flan-T5 **Thrust** achieves better performance with 40% examples rejecting external knowledge usage compared to external knowledge used either on no or all examples. Such observations show the potential of using **Thrust** on current instruction-finetuned models.

## A.5   Ablation on the design choices of *Thrust*

Following [63], we use a few-shot multitask binary NLI dataset to test the influence of each factor of **Thrust** (i.e., FS-NLI), through measuring how well the metric and its variants can measure the with the hardness of a diverse set of datasets. From Table 5, we can observe that all the design choices are crucial to the success of using **Thrust**  to detect how hard a query is for a given task and model.

Table 5: Compare **_Thrust_** to its various variants following the setting of the original work [63], where higher correlation denotes that the metric can better capture the hardness of tasks with respect to a given model (RoBERTa-large [32]). *without direction* denotes the variant to use scalar instead of vectors for **_Thrust_**. The best-performing entry is marked in bold.

| Metric | Correlation |
|---|---|
| **_Thrust_** | **0.45** |
| without cluster size | 0.23 |
| without direction | 0.19 |
| without distance | 0.06 |
| cosine distance | 0.08 |
| one cluster per label class | 0.32 |
| ten clusters per label class | 0.12 |
| cluster size to inertia | 0.03 |

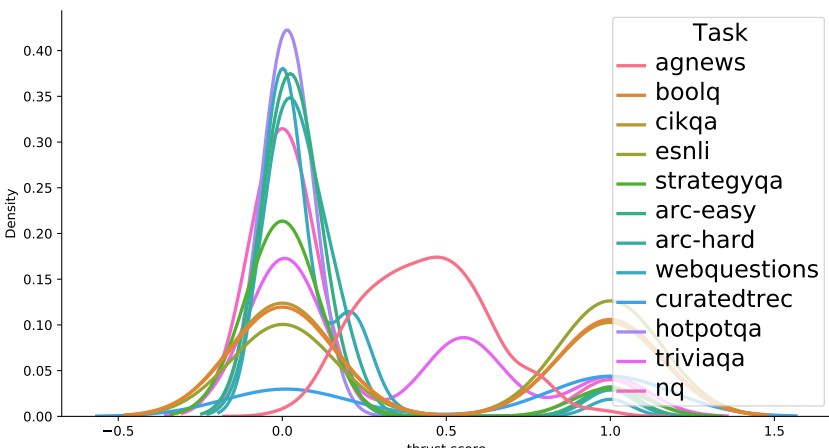

Figure 8: Distribution of **_Thrust_** scores for all involved tasks with UnifiedQA-3b to create the instance representation. The distribution is normalized by Kernel Density Estimation. Low scores denote the cases where internal knowledge is not enough, and vice versa.

## A.6 Full distribution of *Thrust* across tasks

Figure 8 demonstrates the distribution of **_Thrust_** scores for each of the involved tasks. Besides the findings in the main content that **_Thrust_** can help identify the knowledge necessity for various tasks through viewing the distribution, we can also observe that **_Thrust_** can lead to a diverse distribution of scores that may contain multiple peaks.

## A.7 Performance of using external knowledge (in table)

Table 6 presents the performance in Figure 3 of the original submission in a table format. Similarly, we can observe that it is not trivial to use external knowledge, especially in the zero-shot settings, it is possible that models get worse performance with external knowledge, for example, for ARC-C, both T5-base and T5-large show worse performance with the extra knowledge injected. Also, we can observe that external knowledge is crucial for open-domain QA tasks. The gain can be huge, for example, for UnifiedQA-3b, the performance is improved from 18.6 to 80.0, in terms of QA-F1 on TriviaQA, with the external knowledge.

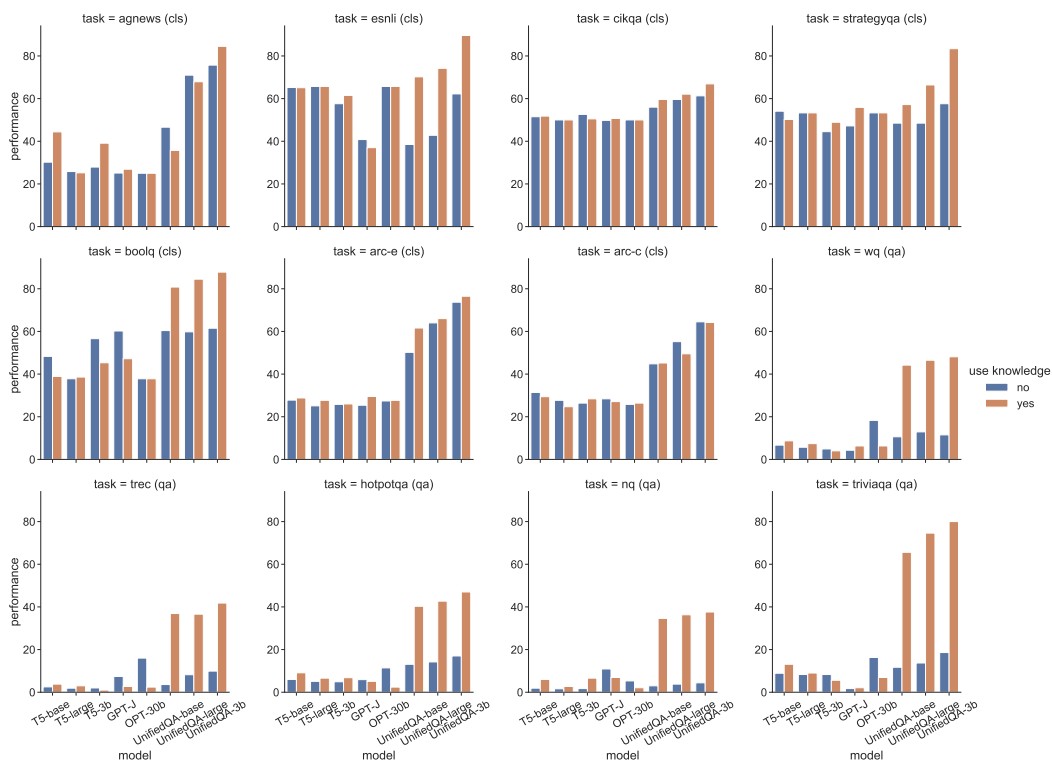

Figure 9: Performance of various models on MC classification tasks (accuracy) and open-domain QA tasks (QA-F1), denoted by (cls) and (qa), respectively. The x-axis represents the model names, which are shared across sub-figures. Use knowledge: yes or no denotes using full knowledge or not for all queries. UnifiedQA denotes T5 models with different sizes fine-tuned on the UnifiedQA dataset.

Table 6: Performance of various models on the MC classification tasks (accuracy) and open-domain QA tasks (QA-F1). Performances without/with knowledge external knowledge are presented before/after the vertical bar, respectively. UnifiedQA-X denotes T5 models with corresponding sizes fine-tuned on the UnifiedQA dataset.

| Model | parameters | AGNews | e-SNLI | CIKQA | StrategyQA | BoolQ | ARC-E | ARC-C |
|---|---|---|---|---|---|---|---|---|
| Zero-shot |
| T5-base | 220M | 30.2 \| 44.4 | 65.2 \| 65.1 | 51.5 \| 51.8 | 54.1 \| 50.2 | 48.3 \| 38.9 | 27.8 \| 28.8 | 31.4 \| 29.4 |
| T5-large | 770M | 25.8 \| 25.2 | 65.7 \| 65.7 | 50.0 \| 50.0 | 53.3 \| 53.3 | 37.8 \| 38.6 | 25.1 \| 27.7 | 27.7 \| 24.7 |
| T5-3b | 3B | 27.9 \| 39.1 | 57.6 \| 61.5 | 52.6 \| 50.5 | 44.5 \| 48.9 | 56.6 \| 45.3 | 25.8 \| 26.0 | 26.4 \| 28.4 |
| GPT-J | 6B | 25.1 \| 26.9 | 40.8 \| 37.0 | 49.8 \| 50.7 | 47.2 \| 55.9 | 60.2 \| 47.2 | 25.4 \| 29.5 | 28.4 \| 27.1 |
| OPT-30b | 30B | 25.0 \| 25.0 | 65.7 \| 65.7 | 50.0 \| 50.0 | 53.3 \| 53.3 | 37.8 \| 37.8 | 27.4 \| 27.7 | 25.8 \| 26.4 |
| Transfer-learning |
| UnifiedQA-base | 220M | 46.6 \| 35.7 | 38.5 \| 70.2 | 56.0 \| 59.6 | 48.5 \| 57.2 | 60.4 \| 80.8 | 50.2 \| 61.6 | 44.8 \| 45.2 |
| UnifiedQA-large | 770M | 71.0 \| 67.9 | 42.8 \| 74.2 | 59.6 \| 62.1 | 48.5 \| 66.4 | 59.8 \| 84.5 | 64.0 \| 66.0 | 55.2 \| 49.5 |
| UnifiedQA-3b | 3B | 75.7 \| 84.5 | 62.2 \| 89.6 | 61.3 \| 66.9 | 57.6 \| 83.4 | 61.5 \| 87.8 | 73.7 \| 76.5 | 64.5 \| 64.2 |

| Model | parameters | | Web Questions | Curated TREC | HotpotQA | NQ | TriviaQA |
|---|---|---|---|---|---|---|---|
| Zero-shot |
| T5-base | 220M | | 6.7 \| 8.7 | 2.5 \| 3.8 | 6.0 \| 9.1 | 1.9 \| 6.0 | 8.9 \| 13.1 |
| T5-large | 770M | | 5.7 \| 7.4 | 1.9 \| 3.0 | 5.1 \| 6.6 | 1.6 \| 2.7 | 8.3 \| 9.0 |
| T5-3b | 3B | | 4.9 \| 4.0 | 2.0 \| 1.0 | 4.9 \| 6.8 | 1.7 \| 6.6 | 8.3 \| 5.6 |
| GPT-J | 6B | | 4.3 \| 6.3 | 7.4 \| 2.7 | 5.9 \| 5.1 | 10.9 \| 6.9 | 1.7 \| 2.1 |
| OPT-30b | 30B | | 18.3 \| 6.3 | 16.0 \| 2.4 | 11.4 \| 2.4 | 5.3 \| 2.1 | 16.3 \| 6.9 |
| Transfer-learning |
| UnifiedQA-base | 220M | | 10.6 \| 44.2 | 3.6 \| 36.9 | 13.1 \| 40.3 | 3.0 \| 34.6 | 11.7 \| 65.6 |
| UnifiedQA-large | 770M | | 12.9 \| 46.5 | 8.2 \| 36.6 | 14.2 \| 42.7 | 3.8 \| 36.3 | 13.7 \| 74.6 |
| UnifiedQA-3b | 3B | | 11.5 \| 48.1 | 9.9 \| 41.8 | 17.0 \| 47.0 | 4.4 \| 37.6 | 18.6 \| 80.0 |

