# OpenReview forum: "Thrust: Adaptively Propels Large Language Models with External Knowledge"
_NeurIPS.cc/2023/Conference — NeurIPS 2023 poster_

### Official Review · Reviewer_LAud · 2023-07-02

**Soundness:** 3 good
**Presentation:** 3 good
**Contribution:** 3 good
**Rating:** 6
**Confidence:** 4

**Summary:**

LLM’s parametric memory may be inaccurate or outdated and thus retrieving additional information to LLMs can help but it’s costly and may be noisy.
So this paper proposes Thrust to measure the instance-level parametric memory and can help determine whether to use a retrieval module for enhancing LLMs, which is more cost-efficiency.
By dynamically using external knowledge, the proposed IAPEK can achieve consistent gains over a large amount of tasks and LLMs.


**Strengths:**

1. The task is important and motivation is well elaborated.
2. Thrust is quite data-efficient to construct, which is valuable in practice.
3. Comprehensive evaluations are conducted and valid points are made.


**Weaknesses:**

1. LLMs used in this paper are not that strong and may not well utilize external knowledge, as pointed by you. I would recommend adding models like FLAN-T5 or other instruction-tuned models that can follow instructions to use external information for QA.

**Questions:**

No questions. This paper is well done.

**Limitations:**

1. One assumption of Thrust is that if PTLM has mastered sufficient knowledge for a task, the hidden states could be used for clustering. But whether using external knowledge is not just about if LLMs can use/represent well. For example, some time-sensitive questions like `who is the CEO of Twitter` can be confidently but wrongly answered by LLMs. However, in these cases, LLMs should use external knowledge.

2. Missing related work:
Xie et al., *Adaptive Chameleon or Stubborn Sloth*: Unraveling the Behavior of Large Language Models in Knowledge Conflicts.
This paper has some conclusions/observations aligned with Thrust: external evidence may mislead the LLMs to generate wrong answers.
Given this paper was available after NeurIPS submission deadline, missing it in reference will not negatively affect my evaluation of Thrust. But you may consider adding it in the next version.

Minor
3. In appendix, Table 3 should be annotated with red and green like you did to the table in the main paper.

---

> ### Author Rebuttal · Authors · 2023-08-09
>
> Thanks for the valuable comments and suggestions. We would like to address your concerns as follows.
>
> **LLM ability (W1)**: Thanks for pointing out the discussion regarding the choice of models. We will try to add Flan-T5 in the camera-ready version to compare if instruction fine-tuning also helps improve external knowledge utilizability.
>
>  Our reason to use the original pre-trained T5 is that UnifiedQA is fine-tuned on these models so that we can compare (in Figure 4) how strongly the model can utilize external knowledge after fine-tuning on this target and show the fine-tuning step in UnifiedQA does help.
>
> On the other hand, UnifiedQA is still comparatively strong on our set of tasks (given the size). For example, LLAMA2 7B achieves 77.4, 75.2, and 45.9 on BoolQ, ARC-e, and ARC-c, respectively. UnifiedQA 3B achieves 87.8, 73.7, and 64.5 in similar settings (Table 20 in LLAMA 2 paper (https://ai.meta.com/research/publications/llama-2-open-foundation-and-fine-tuned-chat-models/)).
>
> **Time Sensitivity (L1)**: Thanks for mentioning this point. We regard time sensitivity as a part that can be done in the IAPEK framework, but not by Thrust, as the framework is motivated by both noise and staticity. We will bring this up in the future work discussion in the final version: another orthogonal sort of score measuring time sensitivity can be designed to decide if updated knowledge retrieval is necessary, for example, based on (Ning. et al., 2022).
>
> **Missing Reference (L2)**: We appreciate your introduction to this great paper. We will add this paper to our discussion regarding the misleading behavior of LLMs in the final version.
>
> **Presentation (L minor 3)**: We appreciate your check on our appendix. We will refine Table 3 to make it consistent with the main paper.
>
>
> **References**
>
> Qiang Ning, Ben Zhou, Hao WU, Haoruo Peng, Chuchu Fan, Matt Gardner. 2022. A Meta-framework for Spatiotemporal Quantity Extraction from Text. In Proceedings of ACL.

---

> > ### Comment · Reviewer_LAud · 2023-08-20
> > **Thanks for Your Response**
> >
> > Thanks for your detailed response and your willingness to continue improving this paper.
> >
> > 1. I agree that UnifiedQA is a strong baseline built upon T5, and I am looking forward to seeing the additional results included in the camera-ready version.
> >
> > 2. It is reasonable to categorize questions as Time Sensitivity and not include them in this paper. The current submission already provides sufficiently comprehensive results.
> >
> > Thank you once again for your reply. I am looking forward to the updated version and your future work on time sensitivity!

---

> > > ### Author Response · Authors · 2023-08-20
> > > **Thank you**
> > >
> > > Thank you so much on your comments! We will discuss more on the time sensitivity and work on adding Flan-T5 experiments in the camera-ready version.

---

### Official Review · Reviewer_RnNH · 2023-07-04

**Soundness:** 3 good
**Presentation:** 3 good
**Contribution:** 2 fair
**Rating:** 5
**Confidence:** 4

**Summary:**

This work proposes methods IAPEK and Thrust to make instance-level decisions about when to utilize external knowledge for question answering. IAPEK is instance-level adaptive propulsion of external knowledge, essentially the use of external knowledge only when it is necessary beyond the base model. Thrust is a heuristic scoring method to decide which questions to use external knowledge on, based on various notions of distance from existing clusters of points. Intuitions for Thrust score are outlined in S2.2. The work demonstrates that Thrust outperforms two baseline methods (random and BM25) at overall accuracy when used to select which instances require external knowledge under a fixed budget. They also demonstrate that using IAPEK with Thrust performs nearly as well as using external knowledge in every case, with a lower computational budget.

**Strengths:**

- The work makes useful statements about the complex nature of retrieval based QA -- both in terms of efficiency and the counterintuitive finding that external knowledge does not always help
- Thrust seems to outperform baselines at selecting instances that require external knowledge
- In general, IAPEK seems to improve efficiency without too much loss of performance compared to always using external knowledge

**Weaknesses:**

- The Thrust score does not have theoretical grounding, or sufficient ablations to justify design decisions. These are not both necessary, but more justification (either theoretical or experimental) for specific design decisions would be very useful. A description of intuitions is given at line 105, but no experimental examples are given to demonstrate that these cases are relevant. One useful aspect would be comparing to simpler scores. Only random and BM25 are used as baselines, but what about simpler notions of distance, that either do not involve complex clustering, or do not involve the "mean vector" idea for Thrust? Something simpler like distance from existing points may perform worse, but there it is not clear given the current results included in the paper. Perhaps some of the complexity of Thrust is not required.
- More broadly, it would be useful to show, even for a subset of datasets, the full set of possibilities between: {no EK, IAPEK default, BM25, Thrust, distance from overall centroid, full EK}
- It is not completely clear what aspects factor into the performance of IAPEK/Thrust. Particularly, the authors mention that external knowledge can sometimes hurt performance, in which case perhaps Thrust is helping less with efficiency, and more with preventing such examples from seeing external knowledge that would reduce performance. One useful aspect would be a more comprehensive version of Table 3 with numbers, to answer the question "how often does thrust outperform full EK?"
- Continuing from the point above, it is not clear from the paper whether the justification of Thrust is improving efficiency (i.e. full EK always helps but is expensive) or performance (i.e. there are some examples that do better without EK, and Thrust helps identify these)

**Questions:**

Question: have you considered selecting examples based on notions of uncertainty? e.g. just looking at the entropy of the model on the output space, rather than interacting directly with vectors or model internals. It would seem like a less certain model may make more use of external knowledge.
See weaknesses for some questions.

**Limitations:**

It would be useful to include more information about limitations. What is the maximally useful case you see for your work, and what will still be left to do in solving this problem?

---

> ### Author Rebuttal · Authors · 2023-08-09
>
> Thanks for the valuable comments and suggestions.  For your reference, our design choice ablation and limitation discussion are provided in Appendix. We would like to address your concerns as follows.
>
> **Design Choice (W1)**: We kindly refer you to Appendix (A.4 and Table 2) included in the supplementary materials. As discussed in Footnote 4 of page 5, we presented our experiments on design choice ablation and known limitations in the appendix. We compared the alternative choices such as no cluster size, no direction, using inertia, etc, to justify our design of Thrust. Thanks for your suggestions and we will include this section in the main paper in the final version.
>
> **Results Comparison (W2)**: Thanks for your suggestions on the presentation. Our experiments contain the full set of datasets for {no EK, IAPEK default, IAPEK BM25, Thrust, Full EK}. {no EK, Full EK} are compared in Figure 4. {IAPEK default, BM25, Thrust} are compared in Table 1 and Table 2. These experiments are all conducted on the same set of datasets. We will include the table containing all the entries in Appendix in our final version.
>
> **Details about Performance vs. Efficiency (W3)**: Thanks for your suggestions for the table design. In Table 3, we present that in ⅔ cases, Thrust outperforms 99% of full EK performance when saving at least 10% expense.
>
> More details are as follows: in ⅓ -  ½  cases, Thrust rejects external noise and outperforms full EK. For datasets such as BoolQ and CIKQA, where the external knowledge can be noisy, Thrust identifies “some examples that do better without EK” and always outperforms full EK. For datasets such as e-SNLI, where humans annotated the external knowledge,  full EK always helps but is expensive. The specific datasets are as follows:
>
> For UnifiedQA-base
>
> Thrust > full EK: BoolQ, CIKQA, StrategyQA, ARC-C, TriviaQA, AGNews
>
> Thrust > 99% full EK: ARC-E. wq. HotpotQA
>
> Thrust < full EK: e-SNLI, TREC, NQ
>
> For UnifiedQA-3b
>
> Thrust > full EK: BoolQ. CIKQA, ARC-C, HotpotQA
>
> Thrust > 99% full EK: StrategyQA, AGNews, ARC-E, TriviaQA, NQ
>
> Thrust < full EK: e-SNLI, WQ, TREC
>
>
> We will add a new table and more discussion comparing Thrust and full EK in the final addition.
>
> One step further, we also identify this comparison as a Performance-Efficiency trade-off with controlling the expected EK rejection rate. Given extra space provided upon acceptance, we will present and analyze the trade-off with an AUC-ROC-like curve, where a convex curve is desired.
>
>
> **Uncertainty (Question)**: The entropy of the model output could be applied to IAPEK for classification tasks. However, some answers are of various lengths for open-domain question answering. An example of NQ is: What does a drink from Narcissus's Spring cause the drinker to do, the expected answer is “fall in love with themselves” rather than from a fixed label set, which can make the entropy test unstable. To provide a unified framework for classification and open-domain QA tasks, we designed Thrust for IAPEK. On the other hand, Thrust can also be regarded as a way to measure uncertainty. We will point this out as potential future work in the final version.
>
> **Limitations**: We kindly refer you to Appendix A.1 contained in the original supplementary materials, where we discuss the potential limitations of our work (e.g., can not be used for black-box LLMs). We will include this section in the main paper in the final version. Thanks again for the suggestions.

---

> > ### Comment · Reviewer_RnNH · 2023-08-22
> >
> > Thank you for your response. I have raised the score to 5.

---

> > > ### Author Response · Authors · 2023-08-22
> > > **Thank you**
> > >
> > > Thank you again for your great comments and suggestions!

---

### Official Review · Reviewer_2iTX · 2023-07-05

**Soundness:** 2 fair
**Presentation:** 2 fair
**Contribution:** 2 fair
**Rating:** 4
**Confidence:** 4

**Summary:**

The paper addresses the limitations of large-scale pre-trained language models (PTLMs) in effectively utilizing external knowledge. It proposes the instance-level adaptive propulsion of external knowledge (IAPEK) as a solution to leverage external knowledge only when necessary. The paper introduces a novel metric called "Thrust" to measure the knowledgeability of PTLM models at the instance level, using the representation distribution of a small number of seen instances. Extensive experiments demonstrate that Thrust is an effective measurement of PTLM models' instance-level knowledgeability. The paper shows that using the Thrust score as a retrieval indicator achieves significantly higher cost-efficiency compared to naive usage of external knowledge, resulting in a 26% average performance improvement on 88% of the evaluated tasks. These findings contribute to the understanding and real-world application of knowledge-enhanced language models, particularly in scenarios with limited knowledge-seeking budgets due to computation latency or costs.






**Strengths:**

1、The authors clearly highlight the limitations of implicit knowledge in pre-trained language models (PTLMs), such as being opaque, static, and inefficient. This acknowledgment sets the stage for proposing a novel approach to address these limitations.

2、The authors introduce the concept of Instance-level Adaptive Propulsion of External Knowledge (IAPEK) as a solution to address the limitations mentioned earlier.

**Weaknesses:**

1、The methods and experiments are described in an informal and obscure manner, lacking motivation for the specific choices made and failing to compare them with alternative approaches. Moreover, this paper lacks the experimental comparison and discussion with ChatGPT and other instruction fine-tuning large models

2.  A major weakness of this work is its lack of reproducibility. The paper fails to provide clarity on whether the external knowledge used is published or not, and it does not explain how one can obtain it.


**Questions:**

How does IAPEK combine with instruction fine-tuning of large models?

**Limitations:**

The authors do not discuss the significance of this work in the context of large models.

---

> ### Author Rebuttal · Authors · 2023-08-09
>
> Thanks for the valuable comments and suggestions. We would like to address your concerns as follows. For your reference, our design choice ablation and reproducibility check are provided in Appendix and referred to in footnotes 4 and 1. Details are as follows:
>
> **Design Choice (W1)**: We kindly refer you to Appendix (A.4 and Table 2) included in the supplementary materials. As said in Footnote 4 of page 5 in our submission, we presented our design choice ablation and known limitations in the appendix. We compared the alternative choices such as no cluster size, no direction, using inertia, etc, and validated our design of Thrust.
>
> **Reproducibility (W2)**: We kindly refer you to Footnote 1 of page 1 of our submission, where we promise that “the code and data(including the external knowledge collected) will be released upon acceptance”. On the other hand, we introduce the details of the data collection in Section 3.1. Tools used, such as DPR and Wikipedia paragraphs, are all publicly available at https://github.com/facebookresearch/DPR or https://github.com/castorini/pyserini. For more details, we also included a sample dataset (sample_dataset.json) in the supplementary material. Thank you.
>
> **Instruction fine-tuning model (Questions)**: We deliberately design Thrust to only rely on the dev and test queries so that the external knowledge can be arbitrary. For example, in the context of the instruction fine-tuning model, we can use Thrust to rank the queries and only conduct Chain-of-thought on the hard examples. On the other hand, we can also use Thrust to suggest if further details of the question are needed to be provided. Thanks for your suggestion on the scope of instruction fine-tuning models, we will suggest this potential usage in discussion in the final version.
>
> **Presentation**: We will improve our presentation on methods and experiments in our final version. Some points are mentioned in our rebuttal to Reviewer eb2S.

---

### Official Review · Reviewer_eb2S · 2023-07-07

**Soundness:** 3 good
**Presentation:** 3 good
**Contribution:** 3 good
**Rating:** 7
**Confidence:** 4

**Summary:**

The authors propose a thrust score which measures if a pre-trained language (PTLM) model has the knowledge to perform the task. They then go on to use this score to choose when they should use external knowledge (when the thrust score is less). The main crux of the thrust score is knowledge representation. The authors argue that if the PTLM places a sample close to related samples, then it has sufficient knowledge about the sample.

I feel the thrust score is very good contribution. The adaptive lookup for knowledge experiments are useful, though there might be other ways to use this thrust score.

**Strengths:**

1. The authors propose a thrust score which is a measure of a pre-trained language models knowledge of the instance. These are based on the distance of a sample to the centroid of a cluster. The clusters are task examples. There can even be mulitple clusters within a class in a classification task. This is measured from representations in the last hidden layer (last decoder layer in T5).

2. Given the authors have a thrust score, they proceed to lookup external knowledge only when the model does not have internal knowledge to predict on a sample. The hypothesis is that looking up external knowledge all the time is wasteful and some times even counter productive because of noisy external knowledge.

3. Overall I think the thrust score is a very useful contribution.

4. The ablations - checking which layer to use for representations, comparison with BM25, comparison with full knowledge usage are meaningful.

**Weaknesses:**

1. The adaptive knowledge injection (while useful and demonstrates that the thrust score is effective) could have benefited with knoweldge probing experiments rather than just QA, MC etc. The performance on (triples) tail prediction task, or even knowledge probing like in the below work, can add be even more interesting.

Onoe, Y., Zhang, M.J., Padmanabhan, S., Durrett, G. and Choi, E., 2023. Can LMs learn new entities from descriptions? Challenges in propagating injected knowledge. arXiv preprint arXiv:2305.01651.

2. Some of the claims like below seem a bit subjective and can be avoided. Atleast from the abstract.

"we can achieve significantly higher cost-efficiency with Thrust score as the retrieval indicator than the naive usage of external knowledge on 88% of the evaluated tasks with 26% average performance improvement."

3. The results could be presented a bit better. For example, figure 4 seems unnecessarily complicated.

It's ofcourse the author prerogative, but it took me a while to get used to the thrust, propulsion terminology.

**Questions:**

1. Can the thrust score be used for knowledge injection using additional pre-training or fine-tuning? Some time ago, there was a paper that used to additionally pre-train on domains where it was performing worse.

2. Have you considered knowledge prompts or pre-fix tuning on downstream tasks besides adding the external knowledge and "Answer:" to the prompt?

There is a typo on line 271. Thrust can help identifies instances requiring

**Limitations:**

There isn't a separate discussion on limitations but they do mention tasks where this approach under-performs baselines.

---

> ### Author Rebuttal · Authors · 2023-08-09
>
> Thanks for the valuable comments and suggestions. We would like to address your concerns as follows.
>
> **Extended Usage (W1 & Q1)**: we agree that adaptive knowledge injection can be extended to other cases such as ECBD or EKP (Onoe et al., 2022, 2023). We will include this line of work in the discussion in the camera-ready version.
> By design, Thrust is independent of the type of external knowledge, so that the adaptively used external knowledge can be any sort, for example, definitions in EKP. Furthermore, Thrust can also be used as a way to measure performance without extensive fine-tuning. In Section A.4 in the appendix included in the supplementary material, we show that Thrust can also be regarded as a dataset hardness metric following the setting of (Zhao et. al., 2022), so that, with an active learning scheme (e.g., Tamkin et. al., 2022), we can use Thrust to rank example hardness and potentially schedule the pre-training or fine-tuning.
>
> **Presentation (W2,3 & Limitations)**:  (1) We will remove the subjective expressions (e.g., significantly) and correct all typos. (2) We will simplify the figure by only presenting the results for a part of the models: T5-3b, GPT-J, OPT-30b, and UnifiedQA-3b. We will put the detailed figure in Appendix. (3) We will change propulsion to augmentation. For the term Thrust, we will consider seeking a better acronym; (4) Our limitation discussion was included in Appendix. We will put them back in the main doc in the final version. We discussed the cold start and black-box LLM problems. Thanks so much for the suggestions!
>
> **Knowledge Prompts (Q2)**: In this paper, we mainly study when we shall add external knowledge and the way to use that knowledge is out of the scope of this work. We mainly follow the setting of the best knowledge utilization model in our experiments (i.e., UnifiedQA).
> Will add the discussion about the potential to incorporate Thrust with knowledge prompts and prefix tuning as an important future direction.
>
> **References**:
>
> Xinran Zhao, Shikhar Murty, and Christopher D. Manning. 2022. On measuring the intrinsic few-shot hardness of datasets. In Proceedings of EMNLP.
>
> Alex Tamkin, Dat Pham Nguyen, Salil Deshpande, Jesse Mu, and Noah Goodman. 2022. Active learning helps pretrained models learn the intended task. In Advances in Neural Information Processing Systems

---

> > ### Comment · Reviewer_eb2S · 2023-08-16
> > **Thanks**
> >
> > Thanks for incorporating the suggestions. Yes, understand the focus of your work is on when to use external knowledge. Was just curious if you tried PEFT methods.

---

> > > ### Author Response · Authors · 2023-08-17
> > > **Thanks**
> > >
> > > Thank you so much for your comments on extending the scope of our work. We will point it out in the discussion and work on how Thrust can cooperate with PEFT methods such as LORA, Prefix-tuning, Soft Prompting, and Adapter. Besides the active learning scheme mentioned, another way can be to use Adapter layer representation to calculate the Thrust score.

---

### Decision · Program_Chairs · 2023-09-21

**Decision:**

Accept (poster)

**Comment:**

This paper tackles an important problem of estimating when an LLM lacks the information needed to answer a given query, triggering the retrieval of new information.  Using this score improves performance compared to naive retrieval by a significant amount in experiments.  Reviewers had some mixed opinions on the submission, with some arguing for acceptance and others on the borderline.  However, the concerns in the most negative review (regarding reproducibility and justification of design decisions) were addressed well in the author response.  One remaining concern is that the paper does not evaluate against many other methods of estimating model confidence, which seems like a potentially significant gap.  Evaluating some of the confidence estimation methods listed in Section 5 of the paper, in addition to the random and BM25 methods that are currently explored, could make the paper's experiments more definitive.